# Identification of a Kitaev quantum spin liquid by magnetic field angle dependence

Kyusung Hwang [1,6], Ara Go [2,3,6], Ji Heon Seong[4], Takasada Shibauchi [5] & Eun-Gook Moon [4✉]

Quantum spin liquids realize massive entanglement and fractional quasiparticles from localized spins, proposed as an avenue for quantum science and technology. In particular, topological quantum computations are suggested in the non-abelian phase of Kitaev quantum spin liquid with Majorana fermions, and detection of Majorana fermions is one of the most outstanding problems in modern condensed matter physics. Here, we propose a concrete way to identify the non-abelian Kitaev quantum spin liquid by magnetic field angle dependence. Topologically protected critical lines exist on a plane of magnetic field angles, and their shapes are determined by microscopic spin interactions. A chirality operator plays a key role in demonstrating microscopic dependences of the critical lines. We also show that the chirality operator can be used to evaluate topological properties of the non-abelian Kitaev quantum spin liquid without relying on Majorana fermion descriptions. Experimental criteria for the non-abelian spin liquid state are provided for future experiments.

[1] School of Physics, Korea Institute for Advanced Study (KIAS), Seoul 02455, Korea. [2] Center for Theoretical Physics of Complex Systems, Institute for Basic Science (IBS), Daejeon 34126, Korea. [3] Department of Physics, Chonnam National University, Gwangju 61186, Korea. [4] Department of Physics, Korea Advanced Institute of Science and Technology (KAIST), Daejeon 34141, Korea. [5] Department of Advanced Materials Science, University of Tokyo, Kashiwa, Chiba 277-8561, Japan. [6]These authors contributed equally: Kyusung Hwang, Ara Go. ✉email: egmoon@kaist.ac.kr

A quantum spin liquid (QSL) is an exotic state of matter characterized by many-body quantum entanglement[1–3]. In contrast to weakly entangled magnetic states, QSLs host emergent fractionalized quasiparticles described by bosonic/fermionic spinons and gauge fields[4,5]. The exactly solvable honeycomb model by Kitaev reveals the exact ground and excited states featured with Majorana fermions and $\mathbb{Z}_2$ gauge fluxes, so-called Kitaev quantum spin liquid (KQSL)[6]. Strong spin-orbit coupled systems with $4d$ and $5d$ atoms such as $\alpha$-RuCl$_3$ are proposed to realize KQSL[7–16], and related spin models have been studied intensively[17–48].

Recent advances in experiments have unveiled characteristics of QSLs. For $\alpha$-RuCl$_3$, signatures of Majorana fermion excitations have been observed in various different experiments of neutron scattering, nuclear magnetic resonance, specific heat, magnetic torque, and thermal conductivity[49–65]. Among them, the half quantization of thermal Hall conductivity $\kappa_{xy}/T = (\pi/12)(k_B^2/\hbar)$ may be interpreted as the hallmark of the presence of Majorana fermions and the non-abelian KQSL[60,63]. At higher magnetic fields, a significant reduction of $\kappa_{xy}/T$ also suggests a topological phase transition[60,66]. Thermal Hall measurements are known to be not only highly sensitive to sample qualities[64] but also very challenging due to the required precision control of heat and magnetic torque from strong magnetic fields. This strongly motivates an independent way to detect the Majorana fermions and non-abelian KQSL.

In this work, we propose that the non-abelian KQSL may be identified by the angle dependent response of the system under applied magnetic fields. As a smoking gun signature of the KQSL, quantum critical lines are demonstrated to occur on a plane of magnetic field directions whose existence is protected by topological properties of the KQSL. The critical lines vary depending on the microscopic spin Hamiltonian, which we show by investigating a chirality operator via exact diagonalization. We further propose that the critical lines can be detected by heat capacity measurements and provide experimental criteria for the non-abelian KQSL applicable to the candidate material $\alpha$-RuCl$_3$.

## Results

**Model and symmetries.** We consider a generic spin-1/2 model on the honeycomb lattice with edge-sharing octahedron crystal structure,

$$H_{KJ\Gamma\Gamma'} = \sum_{\langle jk \rangle_\gamma} \left[ K S_j^\gamma S_k^\gamma + J \mathbf{S}_j \cdot \mathbf{S}_k + \Gamma \left( S_j^\alpha S_k^\beta + S_j^\beta S_k^\alpha \right) \right.$$
$$\left. + \Gamma' \left( S_j^\alpha S_k^\gamma + S_j^\gamma S_k^\alpha + S_j^\beta S_k^\gamma + S_j^\gamma S_k^\beta \right) \right],$$

so-called $K$-$J$-$\Gamma$-$\Gamma'$ model[10,11,13,16]. Nearest neighbor bonds of the model are grouped into $x$, $y$, $z$-bonds depending on the bond direction (Fig. 1a). Spins ($\mathbf{S}_{j,k}$) at each bond are coupled via the Kitaev ($K$), Heisenberg ($J$), and off-diagonal-symmetric ($\Gamma$, $\Gamma'$) interactions. The index $\gamma \in \{x, y, z\}$ denotes the type of bond, and the other two $\alpha, \beta$ are the remaining components in $\{x, y, z\}$ other than $\gamma$. Under an applied magnetic field ($\mathbf{h}$), the Hamiltonian becomes

$$H(\theta, \phi) = H_{KJ\Gamma\Gamma'} - \mathbf{h}(\theta, \phi) \cdot \sum_j \mathbf{S}_j. \quad (1)$$

We specify the magnetic field direction with the polar and azimuthal angles $(\theta, \phi)$ as defined in Fig. 1b. $H_{KJ\Gamma\Gamma'}$ possesses the symmetries of time reversal, spatial inversion, $C_3$ rotation about the normal axis to each hexagon plaquette, and $C_2$ rotation about each bond axis (Fig. 1a). The $C_3$ and $C_2$ rotations form a dihedral group $D_3$. Under each of these symmetries, $H(\theta, \phi)$ is transformed to $H(\theta', \phi')$ with a rotated magnetic field $\mathbf{h}(\theta', \phi')$; see Supplementary Notes 1 and 2.

In the pure Kitaev model, parton approach provides the exact wave function of KQSL together with gapped $\mathbb{Z}_2$ flux and gapless Majorana fermion excitations. Application of magnetic fields drives the KQSL into the non-abelian phase by opening an energy gap in Majorana fermion excitations. The gap size is proportional to the mass function, $M(\mathbf{h}) = h_x h_y h_z / K^2$, and the topological invariant (Chern number) of the KQSL is given by the sign of the mass function, sgn($h_x h_y h_z$)[6].

**Topologically protected critical lines.** Topological invariant of non-abelian phases with Majorana fermions can be defined from the quantized thermal Hall conductivity, $\kappa_{xy}/T = \nu(\pi/12)(k_B^2/\hbar)$, where $\nu$ is the topological invariant representing the total number of chiral Majorana edge modes ($T$: temperature)[6]. While the topological invariant in the pure Kitaev model is exactly calculated by the Chern number of Majorana fermions, it is a nontrivial task to analyze the topological invariant for the generic model $H(\theta, \phi)$.

Our strategy to overcome the difficulty is to exploit symmetry properties of the topological invariant and find characteristic features of the non-abelian KQSL. Concretely, we focus on the landscape of $\nu(\mathbf{h})$ on the plane of the magnetic field angles $(\theta, \phi)$. Our major finding is that critical lines of $\nu(\mathbf{h})$ must arise as an intrinsic topological property of the non-abelian KQSL.

We first consider time reversal symmetry and note the following three facts:

- Time reversal operation reverses the topological invariant as $\nu(\mathbf{h}) \rightarrow -\nu(\mathbf{h})$.
- Time reversal operation also reverses the magnetic field direction: $\mathbf{h}(\theta, \phi) \rightarrow -\mathbf{h}(\theta, \phi) = \mathbf{h}(\pi - \theta, \phi + \pi)$.
- Topologically distinct regions with $\{+\nu(\mathbf{h}), +\mathbf{h}\}$ and $\{-\nu(\mathbf{h}), -\mathbf{h}\}$ exist on the $(\theta, \phi)$ plane.

These properties enforce the two regions to meet by hosting critical lines where Majorana fermion excitations become gapless. In other words, topological phase transitions must occur as the field direction changes. We propose that the very existence of critical lines can be used in experiments as an identifier for the KQSL.

We further utilize the $D_3$ symmetry of the system. The topological invariant $\nu$ and thermal Hall conductivity $\kappa_{xy}$ are $A_2$ representations of the $D_3$ group, i.e., even under $C_3$ rotations but odd under $C_2$ rotations, which reveals the generic form,

$$\nu(\mathbf{h}) = \text{sgn}[\Lambda_1 (h_x + h_y + h_z) + \Lambda_3 h_x h_y h_z], \quad (2)$$

where $\Lambda_{1,3}$ are field-independent coefficients. The $h$-linear term ($h_x + h_y + h_z$) and $h$-cubic term ($h_x h_y h_z$) are the leading order $A_2$ representations of magnetic fields. Conducting third order perturbation theory, we find the coefficients

$$\Lambda_1 = -\frac{4\Gamma'}{\Delta_{\text{flux}}} + \frac{6J\Gamma'}{\Delta_{\text{flux}}^2} - \frac{4\Gamma\Gamma'}{\Delta_{\text{flux}}^2} + \frac{5\Gamma'^2}{2\Delta_{\text{flux}}^2} \;\&\; \Lambda_3 = \frac{18}{\Delta_{\text{flux}}^2}, \quad (3)$$

where $\Delta_{\text{flux}} = 0.065|K|$ means the flux gap in the Kitaev limit. See Supplementary Notes 3–5 and ref. [67] for more details of the perturbation theory.

Notice that the $h$-linear term is completely absent in the pure Kitaev model ($\Lambda_1 = 0$). Figure 1c visualizes the topological invariant, $\nu(\mathbf{h}) = \text{sgn}(h_x h_y h_z)$. The dashed lines highlight the critical lines representing the topological phase transitions between the phases with $\nu(\mathbf{h}) = \pm 1$, where the energy gap of Majorana fermion excitations is closed: $\Delta(\mathbf{h}) = 0$.

Exploiting the symmetry analysis, we stress two universal properties of the KQSL with the $D_3$ symmetry.

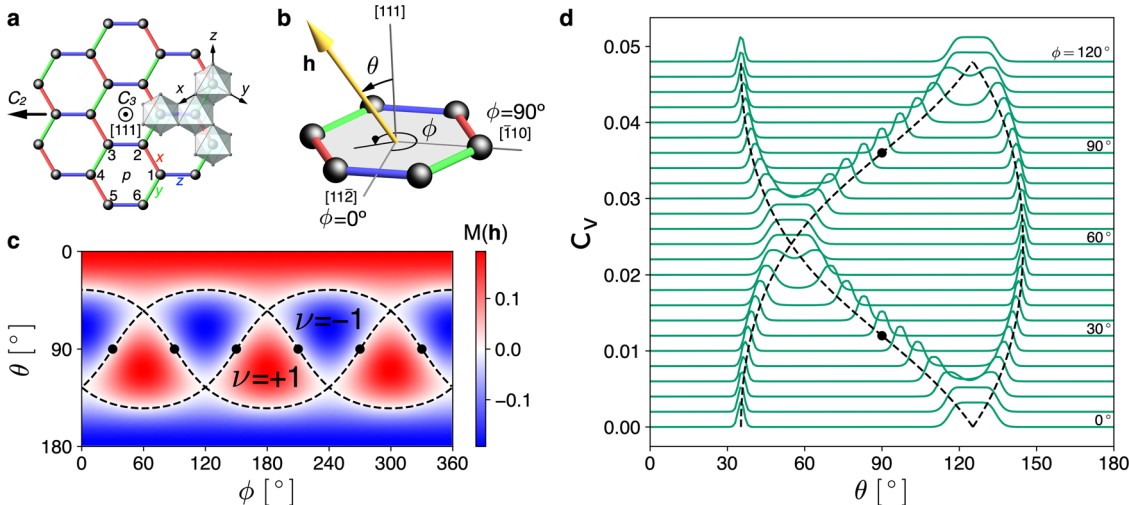

**Fig. 1 Field angle dependence of the pure Kitaev model. a** Honeycomb lattice enclosed by edge-sharing octahedra. Red, green, blue lines denote the $x,y,z$-bonds, and the six numbers indicate the numbering convention for sites in each hexagon plaquette ($p$). Black arrows depict $C_3$ and $C_2$ rotation axes. **b** Convention for the angular representation of an applied magnetic field $\mathbf{h}$ (yellow). The polar and azimuthal angles ($\theta$, $\phi$) are measured from the out-of-plane [111] axis and the bond-perpendicular [11$\bar{2}$] axis, respectively. **c** Color map of the mass function $M(\mathbf{h}) = h_x h_y h_z / K^2$ on the ($\theta$, $\phi$) plane. The dashed lines highlight the topological phase transitions between the $\nu = \pm1$ states, i.e., the quantum critical lines of the energy gap $\Delta(\mathbf{h}) \propto |M(\mathbf{h})| = 0$. The black dots mark the bond directions ($\theta = 90°$, $\phi = 30° + n\cdot60°$), where $n = 0, 1, ..., 5$. **d** A schematic of the field angle dependence of the heat capacity $c_v$ at a fixed temperature, where peak positions determine the critical lines of the non-abelian KQSL.

1. Symmetric zeroes: for bond direction fields, topological transition/gap closing is guaranteed to occur by the symmetry, i.e., $\Delta(\mathbf{h}) = 0$ for ($\theta = 90°$, $\phi = 30° + n\cdot60°$).
2. Cubic dependence: for in-plane fields, the $h$-cubic term governs low field behaviors of the KQSL, e.g., $\nu(\mathbf{h}) \sim \mathrm{sgn}(h_x h_y h_z)$ & $\Delta(\mathbf{h}) \sim |h_x h_y h_z|$ for $\theta = 90°$.

The universal properties and critical lines of the KQSL are numerically investigated for the generic Hamiltonian $H(\theta, \phi)$ in the rest of the paper.

**Chirality operator.** We introduce the chirality operator

$$\hat{\chi}_p = S_2^x S_1^z S_6^y + S_5^x S_4^z S_3^y + C_3 \text{ rotated terms} \qquad (4)$$

at each hexagon plaquette $p$ and investigate the expectation value of the chirality operator, shortly the chirality,

$$\chi(\mathbf{h}) \equiv \frac{1}{N} \sum_p \langle \Psi_{\mathrm{KQSL}}(\mathbf{h}) | \hat{\chi}_p | \Psi_{\mathrm{KQSL}}(\mathbf{h}) \rangle, \qquad (5)$$

and its sign,

$$\bar{\nu}(\mathbf{h}) \equiv \mathrm{sgn}[\chi(\mathbf{h})], \qquad (6)$$

where $|\Psi_{\mathrm{KQSL}}(\mathbf{h})\rangle$ is the ground state of the full Hamiltonian $H(\theta, \phi)$ in the KQSL phase ($N$ is the number of unit cells). The chirality operator $\hat{\chi}_p$ produces the mass term of Majorana fermions and determines the topological invariant in the pure Kitaev limit[6]. More precisely, how magnetic fields couple to the chirality operator determines the topological invariant and the Majorana energy gap. The chirality $\chi$ and its sign $\bar{\nu}$ are in the $A_2$ representation of the $D_3$ group as of the topological invariant $\nu$. We note that the relation between the chirality and the Majorana energy gap, $\chi \sim \Delta$, holds near the symmetric zeroes in a generic KQSL beyond the pure Kitaev model due to the symmetry properties (Supplementary Note 6).

**Exact diagonalization.** The Hamiltonian $H(\theta, \phi)$ is solved via exact diagonalization (ED) on a 24-site cluster with sixfold rotation symmetry and a periodic boundary condition (Fig. 1a). Resulting phase diagrams are provided in the section of **Methods**.

**Table 1 Parameter sets for exact diagonalization and spin wave calculations.**

| Case | $K$ | $J$ | $\Gamma$ | $\Gamma'$ | Phase | Figures |
|---|---|---|---|---|---|---|
| #1 | −1 | 0 | 0 | 0 | KQSL | 2a, 3a |
| #2 | −1 | 0.05 | 0 | 0 | KQSL | 2b, 3b |
| #3 | −1 | 0.05 | 0 | 0.05 | KQSL | 2c, 3c |
| #4 | −1 | 0.08 | 0.01 | 0.05 | KQSL | 2d, 3d |
| #5 | −1 | −0.2 | −0.2 | 0.05 | FM | 4b |
| #6 | −1 | 0.2 | 0.05 | 0.05 | Stripy | 4c |
| #7 | −1 | 0.2 | −0.2 | 0.05 | Vortex | 4d |
| #8 | 1 | 0.2 | −0.2 | −0.05 | Neel | 4e |
| #9 | −1 | −0.3 | 1 | −0.1 | Zigzag | 4f |

Figures 2 and 3 display our major results, the ED calculations of the chirality $\chi_{\mathrm{ED}}(\mathbf{h})$ for the KQSL. The used parameter sets are listed in Table 1. The zero lines [$\chi_{\mathrm{ED}}(\mathbf{h}) = 0$; dashed lines in the figures] exist in all the cases.

The two universal features of the KQSL are well captured by the chirality (Figs. 2 and 3). Firstly, the zero lines of the chirality $\chi_{\mathrm{ED}}(\mathbf{h})$ always pass through the bond directions (marked by black dots), i.e., the symmetric zeroes. Secondly, the chirality shows the cubic dependence for in-plane fields ($\theta = 90°$). The linear term, $h_x + h_y + h_z$, vanishes for in-plane fields, and the cubic term, $h_x h_y h_z$, determines the chirality at low fields, which is confirmed in our ED calculations (lower panels of Fig. 2). Below, we show how non-Kitaev interactions affect topological properties of the non-abelian KQSL.

Most of all, we find that $\bar{\nu}$ becomes identical to $\nu$ for the pure Kitaev model in Fig. 2a. It is remarkable that the two different methods, ED calculations of the chirality sign and the parton analysis, show the complete agreement: $\bar{\nu}(\mathbf{h}) = \nu(\mathbf{h})$. The agreement indicates that the topological phase transitions can be identified by using the chirality operator, which becomes a sanity check of our strategy to employ the chirality operator.

Figure 2b illustrates effects of the Heisenberg interaction ($J$) on the chirality. The shape of the critical lines is unaffected by the Heisenberg interaction, remaining the same as in the pure Kitaev

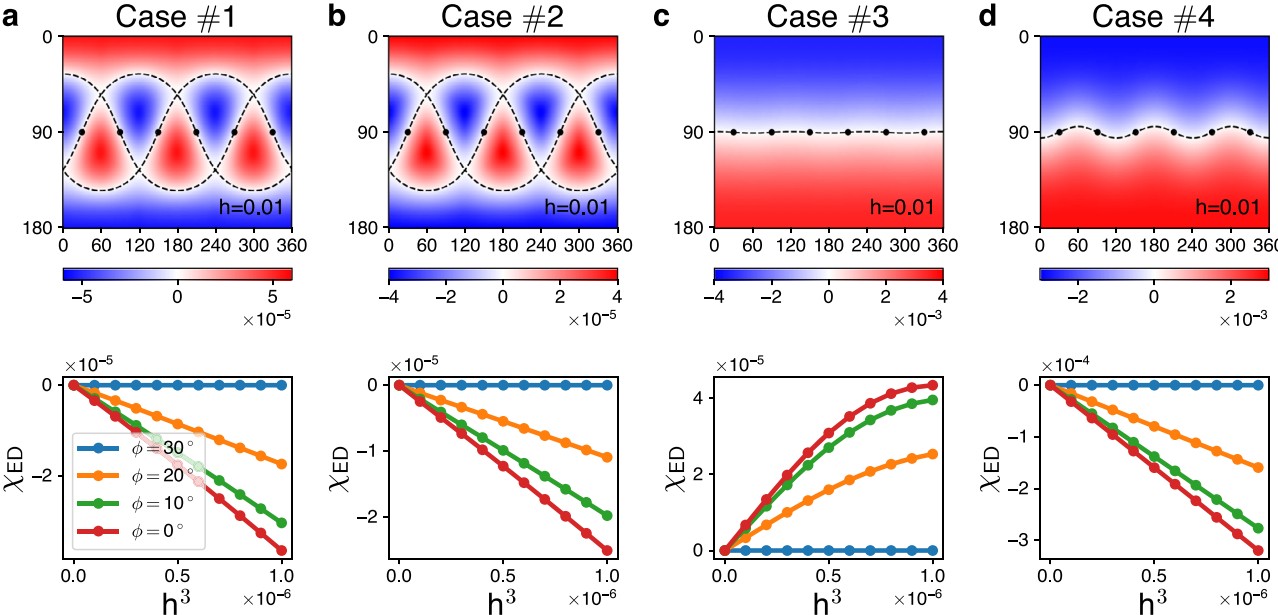

**Fig. 2 Chirality of the non-abelian KQSL.** Upper: color maps of the chirality $\chi_{ED}(\mathbf{h})$ on the plane of the field angles $(\theta, \phi)$, where the magnetic field strength is fixed by $h = 0.01$ (horizontal axis: $\phi[°]$, vertical axis: $\theta[°]$). The dashed lines highlight the zero lines $\chi_{ED}(\mathbf{h}) = 0$, and the black dots mark the bond directions. Lower: $\chi_{ED}(\mathbf{h})$ as a function of $h^3$ for the in-plane fields ($\theta = 90°$, $\phi = 0°, 10°, 20°, 30°$), illustrating the universality of the $h^3$ behavior in the KQSL. In the case #3, the bending at $h^3 > 0.5 \times 10^{-6}$ is an effect of higher order contributions ($h^5, h^7, ...$). The parameter sets used in the four cases (#1~4) are listed in Table 1.

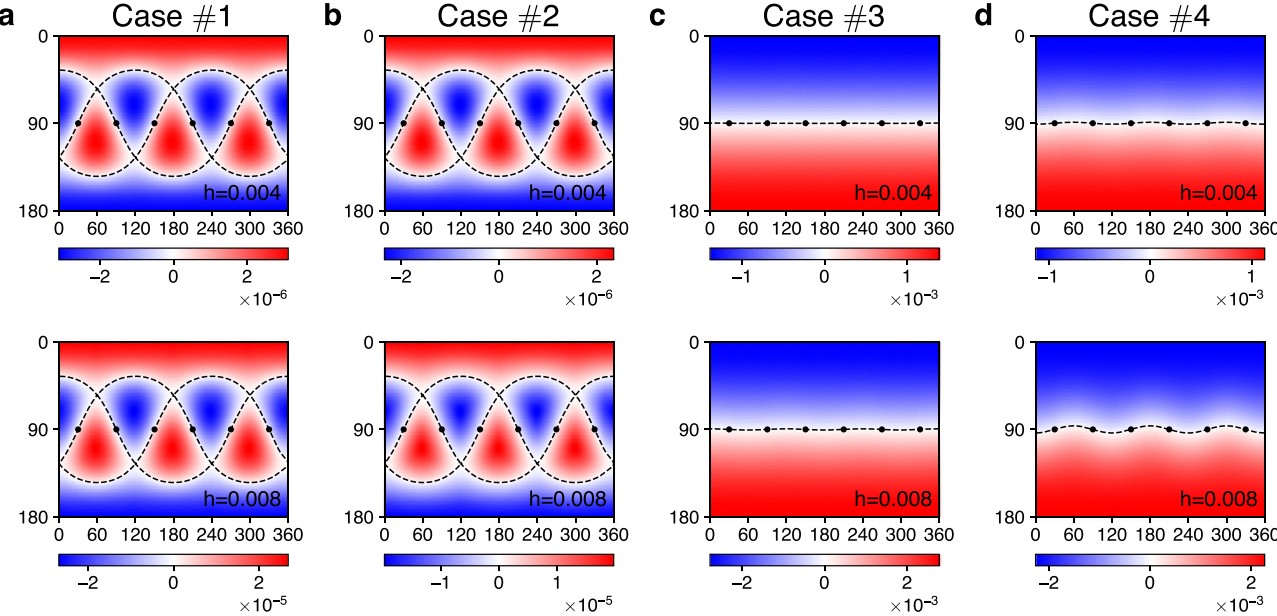

**Fig. 3 Field evolutions of the zero lines.** Color maps of the chirality $\chi_{ED}(\mathbf{h})$ on the plane of the field angles $(\theta, \phi)$ with increasing magnetic field $h = 0.004$, 0.008 (horizontal axis: $\phi[°]$, vertical axis: $\theta[°]$). The dashed lines highlight the zero lines $\chi_{ED}(\mathbf{h}) = 0$, and the black dots mark the bond directions. The parameter sets used in the four cases (#1~4) are listed in Table 1.

model. This result is completely consistent with the perturbative parton analysis [Eq. (3) and Supplementary Fig. 2b], indicating the validity of our strategy.

Figure 2c, d presents consequences of the other non-Kitaev interactions, $\Gamma'$ and $\Gamma$. The zero lines tend to be flatten around the equator $\theta = 90°$, which can be attributed to the $h$-linear term induced by the non-Kitaev interactions: $h_x + h_y + h_z = h\sqrt{3}\cos\theta$. In other words, the zero lines substantially deviate from those of the pure Kitaev model by the non-Kitaev couplings, $\Gamma'$ and $\Gamma$. We point out that the signs of

the chirality are opposite to the Chern numbers of the third order perturbation parton analysis (Supplementary Fig. 2c, d). The opposite signs indicate that the two methods have their own valid conditions, calling for improved analysis (Supplementary Note 8).

Impacts of the non-Kitaev interactions also manifest in the field evolution of the zero lines (Fig. 3). Without the non-Kitaev couplings, $\Gamma'$ and $\Gamma$, the shape of the critical lines is governed by the cubic term $h_x h_y h_z$, as shown in Fig. 3a, b. In presence of the $\Gamma'$ or $\Gamma$ coupling, the $h$-cubic term competes with the $h$-linear term as illustrated in Fig. 3c, d. Namely, the linear term dominates over

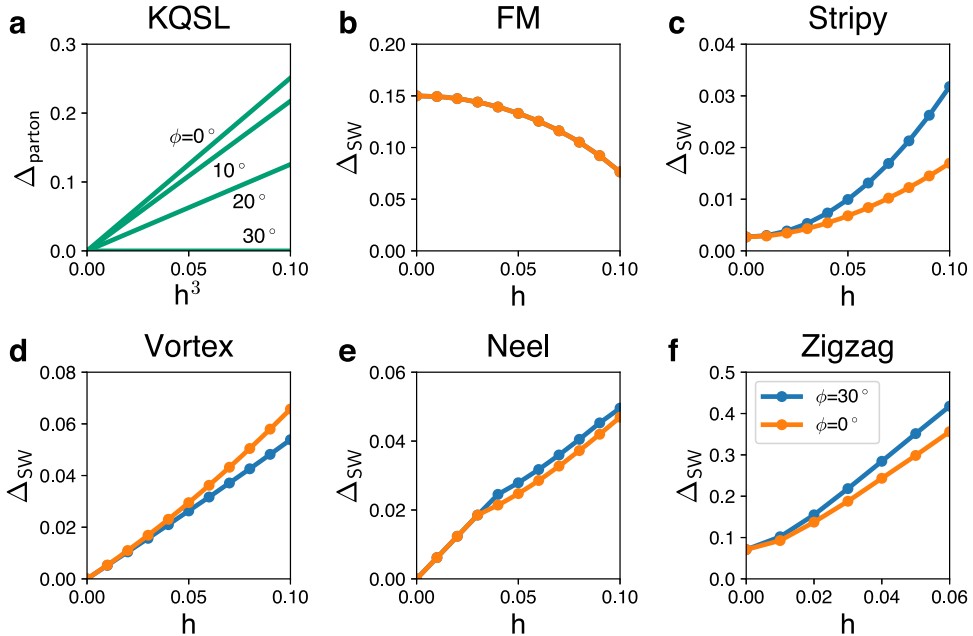

**Fig. 4 Comparison of the KQSL with magnetically ordered phases. a** KQSL: Majorana energy gap $\Delta_{parton}$ as a function of $h^3$ for the in-plane fields, ($\theta = 90°$, $\phi = 0°$, $10°$, $20°$, $30°$), obtained by a parton theory. **b–f** Ferromagnetic (FM), stripy, vortex, Neel, and zigzag phases: Magnon gap $\Delta_{SW}$ as a function of $h$ for the in-plane fields, ($\theta = 90°$, $\phi = 0°$, $30°$), obtained with a spin wave theory for the parameter sets #5~9 in Table 1.

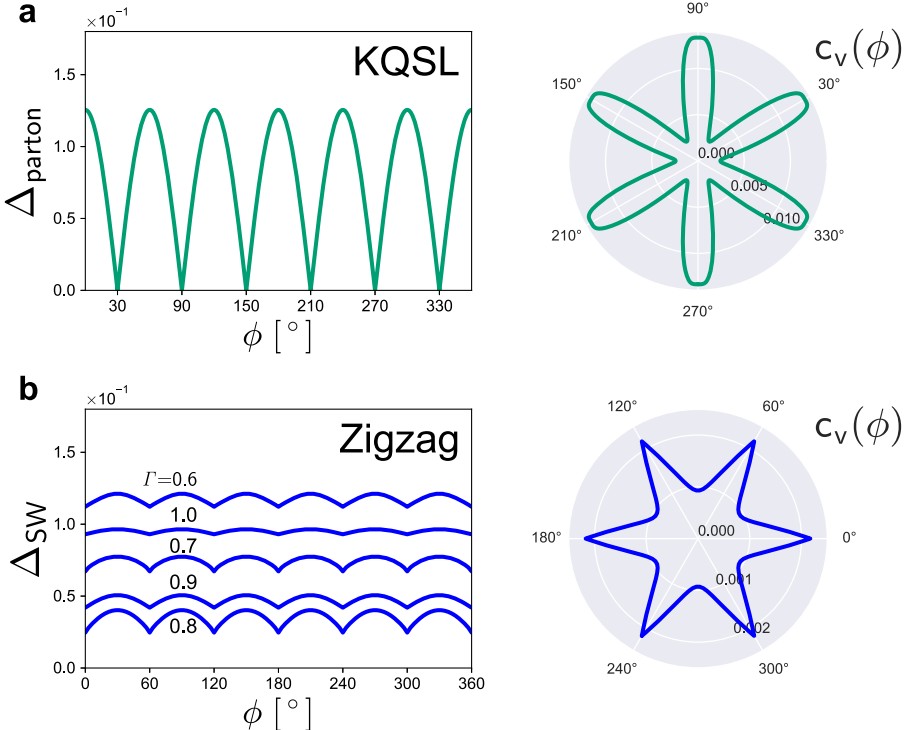

**Fig. 5 Magnetic field angle dependence of specific heat in the KQSL and zigzag states. a** KQSL state: Majorana gap $\Delta_{parton}$ and specific heat $c_v$ as functions of in-plane field angle $\phi$, obtained by a parton theory. **b** Zigzag state: magnon gap $\Delta_{SW}$ and specific heat $c_v$ as functions of in-plane field angle $\phi$, obtained by a spin wave theory with $(K, \Gamma, \Gamma', h) = (-1, 0.8, -0.05, 0.03)$ & $k_B T = 0.01$. Magnon gaps for other values of $\Gamma$ are shown together to highlight the generality of the field angle dependence.

the cubic term at low fields while the dominance gets reversed at high fields (see Supplementary Note 11 and Supplementary Table 4 for more results). The competing nature may be used to quantitatively characterize the non-Kitaev interactions.

Similarities and differences between the topological invariant, $\nu(\mathbf{h})$, and the sign of the chirality, $\bar{\nu}(\mathbf{h})$, are emphasized. First, the two quantities are identical in the Kitaev limit while they can be generally different by non-Kitaev interactions. Second, the two quantities are in the same representation of the $D_3$ group, so $\bar{\nu}(\mathbf{h})$ and $\nu(\mathbf{h})$ have in common the symmetric zeroes. Third, differences between the two quantities may be understood by considering other possible $A_2$ representation spin operators that

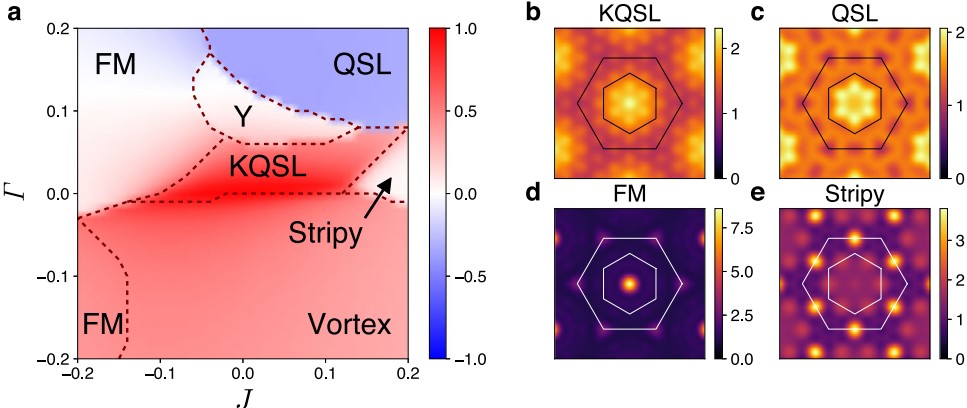

**Fig. 6 Ferromagnetic KQSL and nearby magnetic states. a** Phase diagram of $H_{KJ\Gamma\Gamma'}$ at $K = -1$ and $\Gamma' = 0.05$. The color encodes the flux operator expectation value $\langle \hat{W}_p \rangle$, and the dashed lines denote phase boundaries determined by the ground state energy second derivatives $-\partial^2 E_{gs}/\partial\xi^2 (\xi = J, \Gamma)$. **b–e** Color maps of the spin structure factor $S(\mathbf{q})$ for the KQSL, QSL, ferromagnetic (FM), and stripy states. The inner and outer hexagons denote the first and second Brillouin zones of the honeycomb lattice.

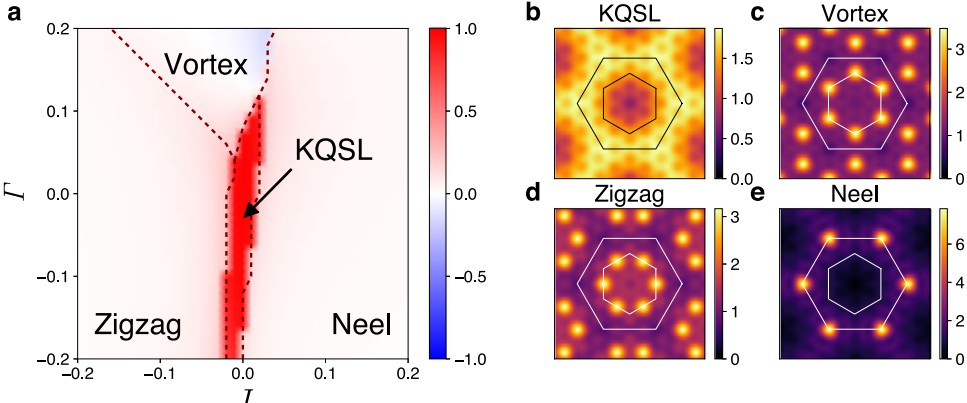

**Fig. 7 Antiferromagnetic KQSL and adjacent magnetic states. a** Phase diagram of $H_{KJ\Gamma\Gamma'}$ at $K = 1$ and $\Gamma' = -0.05$. The color encodes the flux operator expectation value $\langle \hat{W}_p \rangle$, and the dashed lines denote phase boundaries determined by the ground state energy second derivatives $-\partial^2 E_{gs}/\partial\xi^2 (\xi = J, \Gamma)$. **b–e** Color maps of the spin structure factor $S(\mathbf{q})$ for the KQSL, vortex, zigzag, and Neel states. The inner and outer hexagons denote the first and second Brillouin zones of the honeycomb lattice.

may contribute to the topological invariant. For example, linear and higher-order spin operators exist in addition to the chiral operator. Since the topological invariant $\nu$ is related with the thermal Hall conductivity $\kappa_{xy}$, the associated energy current operator directly informs us of relevant spin operators to the topological invariant. We find that linear spin operator is irrelevant to $\kappa_{xy}$ and $\nu$ (Supplementary Note 7). We also evaluate the expectation values of higher-order spin operators for the KQSL and confirm that their sizes are substantially small compared to the chirality (Supplementary Note 8). Therefore, we argue that the critical lines of the non-abelian KQSL are mainly determined by the zero lines of the chirality.

## Discussion

Intrinsic topological properties of the non-abelian KQSL including the critical lines, the symmetric zeroes, and the cubic dependence are highlighted in this work by exploiting the symmetries of time reversal and $D_3$ point group. The chirality operator is used to evaluate the topological properties of the KQSL via the ED calculations. Now we discuss how the properties are affected by lattice symmetry breaking such as stacking faults in real materials. First, the existence of the critical lines relies on time reversal, thus it is not destroyed by lattice symmetry breaking. The symmetric zeroes appearing at the bond directions,

however, are a consequence of the $D_3$ symmetry. The locations of the zeroes get shifted upon breaking the symmetry, which is confirmed in ED calculations of the chirality.

The cubic dependence for in-plane fields also originates from the $D_3$ symmetry and topology in the KQSL. The characteristic nonlinear response is not expected in magnetically ordered phases, which we check by performing spin wave calculations. Figure 4 contrasts the KQSL with magnetically ordered phases in terms of excitation energy gap (Majorana gap vs. magnon gap). The magnetic phases show completely different behaviors from the $h$-cubic dependence. Hence, the characteristic cubic dependence under in-plane magnetic fields may serve as an experimentally measurable signature of the KQSL.

The universal properties of the KQSL can be observed by heat capacity experiments. Figure 5a illustrates the calculated specific heat $c_v(\phi)$ for the KQSL as a function of in-plane field angle $\phi$ (where magnetic field is rotated within the honeycomb plane). The specific heat is maximized by gapless continuum of excitations when the magnetic field is aligned to the bond directions. For comparison, the zigzag state, observed in $\alpha\text{-RuCl}_3$ at zero field, is investigated by using a spin wave theory. The magnon spectrum is gapped due to completely broken spin rotation symmetry, so there is no critical line on the $(\theta, \phi)$ plane (Fig. 5b). Compared to the KQSL, the zigzag state exhibits reverted patterns

of $\phi$ dependence in the excitation energy gap and specific heat. The energy gap is maximized and the specific heat is minimized at the bond directions. This behavior is closely related with the structure of spin configuration: all spin moments are aligned perpendicular to a certain bond direction selected by magnetic field direction (Supplementary Note 9). The distinct patterns of $\phi$ dependence in Fig. 5 characterize differences between the non-abelian KQSL and zigzag states. Remarkably, such behaviors were observed in the recent heat capacity experiments with in-plane magnetic fields[65]. Covering the polar angle ($\theta$) in the heat capacity measurements will provide more detailed information on the critical lines and spin interactions in $\alpha$-RuCl$_3$ (see Fig. 1d).

Lastly, we have examined the chirality and critical behaviors of excitation energy gap for magnetically ordered phases of $H(\theta, \phi)$. It is found that the associated magnon gap does not have any critical lines, and there is no resemblance/correlation between the magnon gap and the chirality (Supplementary Note 9).

To summarize, we have uncovered characteristics of the non-abelian Kitaev quantum spin liquid, including the topologically protected critical lines, the symmetric zeroes, and the $h$-cubic dependence for in-plane fields, by using ED calculations with the chirality operator. Furthermore, we characterize the topological fingerprints of the KQSL in heat capacity. We expect our findings to be useful guides for identifying the KQSL in candidate materials such as $\alpha$-RuCl$_3$. Investigation of the universal properties in field angle dependence of thermodynamic quantities such as spin susceptibility is highly desired, and it would be also useful to apply our results to the recently studied field angle dependence of thermodynamic quantities[61,65,68–71].

## Methods

**Exact diagonalization**. The KQSL and other magnetic phases of $H(\theta, \phi)$ are mapped out by using the flux operator $\hat{W}_p = 2^6 S_1^z S_2^y S_3^x S_4^z S_5^y S_6^x$, the second derivative of the ground state energy $\partial^2 E_{gs}/\partial \xi^2$ ($\xi = J, \Gamma, \Gamma'$), and the spin structure factor $S(\mathbf{q}) = \frac{1}{N}\sum_{i,j}\langle \mathbf{S}_i \cdot \mathbf{S}_j \rangle e^{i\mathbf{q}\cdot(\mathbf{r}_i - \mathbf{r}_j)}$. We find that the KQSL differently responds to non-Kitaev interactions depending on the sign of the Kitaev interaction. Furthermore, the non-abelian phase of the KQSL is ensured by checking topological degeneracy and modular $\mathcal{S}$ matrix[6,38,72].

Figures 6 and 7 display the phase diagrams of $H_{KJ\Gamma\Gamma'}$. A different structure of phase diagram is found depending on the sign of the Kitaev interaction. With the ferromagnetic Kitaev coupling ($K < 0$ as in Fig. 6), the KQSL takes an elongated region along the $J$ axis but substantially narrowed along the $\Gamma$ axis, showing the sensitivity to the $\Gamma$ coupling of the ferromagnetic KQSL. Crossover-type continuous transitions are mostly observed among the KQSL and nearby magnetically ordered states such as ferromagnetic, stripy and vortex states[10,20,43]. Nature of the phase Y is unclarified within the finite size calculation. Unlike the aforementioned magnetically ordered states (Fig. 6d, e), the phase Y does not exhibit sharp peaks and periodicity in the spin structure factor, from which the phase is speculated to have an incommensurate spiral order or no magnetic order. It is remarkable that another quantum spin liquid phase, characterized by negative $\langle \hat{W}_p \rangle$, exists in a broad region of the phase diagram (blue region of Fig. 6a)[39]. The QSL and KQSL show similarity in the spin structure factor (Fig. 6b, c). Nonetheless, the QSL as well as the phase Y get suppressed when the sign of $\Gamma'$ is changed to negative. A zigzag antiferromagnetic order instead sets in under negative sign of $\Gamma'$ (Supplementary Fig. 6).

In case of the antiferromagnetic Kitaev coupling ($K > 0$ as in Fig. 7), the KQSL is found to be more sensitive to the Heisenberg coupling rather than the $\Gamma$ coupling, and surrounded by magnetically ordered states such as the vortex, zigzag, and Neel states[10,20,43]. In contrast to the ferromagnetic KQSL case, phase transitions between the antiferromagnetic KQSL and adjacent ordered states are all discontinuous as shown by $\langle \hat{W}_p \rangle$ in Fig. 7a. We also find that the antiferromagnetic KQSL and ferromagnetic KQSL are distinguished by different patterns of spin structure factor (Figs. 6b and 7b). Further phase diagrams for other values of $\Gamma'$ are provided in Supplementary Fig. 6.

We also examine the phase diagrams at weak magnetic fields, and confirm that the overall structures remain the same as the zero-field results. We find that the chirality is useful for the identification of distinct phase boundaries. In some cases, the chirality performs better than the conventionally used flux (Supplementary Note 12).

We ensure the non-abelian KQSL phase by checking the Ising anyon topological order via threefold topological degeneracy[6,73] and modular $\mathcal{S}$

matrix[38,72,74]. As an example, the $\mathcal{S}$ matrix

$$\mathcal{S}_{ED} = \langle \Psi^{MES-I} | \Psi^{MES-II} \rangle$$

$$= \begin{bmatrix} 0.45e^{-i0.08} & 0.53e^{-i0.03} & 0.70e^{i0.04} \\ 0.53e^{-i0.03} & 0.50 & -0.71e^{-i0.01} \\ 0.70e^{i0.04} & -0.71e^{-i0.01} & 0.02e^{-i2.09} \end{bmatrix} \quad (7)$$

$$\approx \begin{bmatrix} 1/2 & 1/2 & 1/\sqrt{2} \\ 1/2 & 1/2 & -1/\sqrt{2} \\ 1/\sqrt{2} & -1/\sqrt{2} & 0 \end{bmatrix}$$

is obtained for the parameter set #4 in Table 1 with the magnetic field fixed along the [111] direction ($\theta = 0°$). See Supplementary Note 10 for the topological degeneracy and modular matrix computation.

## Data availability

The data that support the findings of this study are available from the corresponding author on reasonable request.

## Code availability

The code used to generate the data in this study is available from the corresponding author upon reasonable request.

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

## Acknowledgements

We thank L. Balents, J. H. Han, Y. B. Kim, and Y. Matsuda. for invaluable discussions. We also thank B. H. Kim and H.-Y. Lee. for useful discussions at the early stage of the project. This work was supported by Institute for Basic Science under Grants No. IBS-R024-D1 (AG), Korea Institute for Advanced Study under Grant No. PG071401 & PG071402 (KH), and NRF of Korea under Grant NRF-2021R1C1C1010429 (AG), NRF-2019M3E4A1080411, NRF-2020R1A4A3079707, and NRF-2021R1A2C4001847 (EGM). Work in Japan was supported by a Grant-in-Aid for Scientific Research on innovative areas "Quantum Liquid Crystals" (JP19H05824) from Japan Society for the Promotion of Science (JSPS), and by JST CREST (JPMJCR19T5). We thank Center for Advanced Computation (CAC) at Korea Institute for Advanced Study (KIAS) for providing computing resources for this work.

## Author contributions

E.-G.M. conceived and supervised the project. K.H., J.H.S., and A.G. performed theoretical calculations. K.H. and T.S. contributed to analysis and comparison of theoretical results and experimental data. K.H., A.G., and E.-G.M. prepared the manuscript with the inputs from T.S.

## Competing interests

The authors declare no competing interests.
