## [Peer Review File · Nature Communications]

Identification of a Kitaev Quantum Spin Liquid by Magnetic Field Angle DependenceREVIEWER COMMENTS

Reviewer #1 (Remarks to the Author):

In this manuscript, the authors propose the magnetic field angle dependence of critical behaviors as an alternative identification for the Kitaev quantum spin liquid. The studies are based on the Kitaev model with additional Heisenberg and off-diagonal-symmetric interactions. The angle plane's critical line is topologically protected and intimately related to a chiral operator by symmetry arguments. Results from the parton theory and the exact diagonalization method also provide support for the proposal.

However, this referee believes that the current manuscript shows insufficient scientific rigor and completeness in several aspects and cannot recommend publication:

1. The study does not show the behavior of the real excitation gap, except for the pure Kitaev model. The Δ_{ED} in Fig. 4 is actually the absolute value of χ_{ED} (which is redundant, and the symbol is quite misleading, especially when Fig. 1 with a similar plot scheme features the parton gap - indeed an energy gap). Does the real gap show critical features aligning with the figures on the left? It is also a directly available result in both parton-based perturbation theory and exact diagonalization calculations.

2. What are the scenarios of χ_{ED} and the real gap outside the Kitaev quantum spin liquid region? For instance, the authors illustrate the critical lines in various circumstances in the Kitaev quantum spin liquid region in Fig. 4. However, how do the critical lines behave in other phases? Such contrast is essential to establish the magnetic field angle dependence as a defining characteristic and intrinsic feature.

3. The need to introduce the chiral spin operator is not clear when the gap value's result should be readily available. By symmetry argument, the chiral spin operator's expectation value vanishes where the gap vanishes, yet their proportionality is not guaranteed and only valid near the critical lines. Why forego the gap for a poor man's alternative that is the chiral spin operator?

4. Using the gap value also helps the analysis be more quantitative and throughout, as it determines the temperature threshold the features smear out. Besides, what role does the strength of the magnetic field h play? What is the optimal magnetic field strength? The manuscript only shows results for a select parameter setting at the very end and largely insufficient considering the authors are making experimentally realistic proposals for alternative KQSL features.

In total, this manuscript begins with an interesting idea yet falls short in subsequent justifications and illustrations and is not suitable for academic publication in its current form.

Dr. Yi Zhang

Reviewer #3 (Remarks to the Author):

Hwang et al. present a theoretical study of the ground state of the Kitaev model with inclusion of non-Kitaev terms (Heisenberg J and the so-called Γ and Γ' terms) in the absence and presence of a magnetic field. For that they use parton theory in the region where the non-Kitaev terms can be treated perturbatively, as well as exact diagonalization of 24-site clusters, which allow to extend the study to finite values of J , Γ and Γ' .

The work is specially devoted to analyze the Chern number of the magnetic field induced phases as a function of the magnetic field angle and to the evaluation of such signatures in thermodynamic quantities such as specific heat.

This is a well-done study that presents nice results worth publishing. I'm less convinced that the results are of enough novelty to have the work published as a Nature Communications.

The reasoning is the following:

1) Calculations of the Chern number (Fig. 1) for the Kitaev model under various applied field directions have been discussed previously, see for instance Yokoi et al. arXiv:2001.01899

2) The variety of phases obtained from exact diagonalization for ferromagnetic and antiferromagnetic Kitaev interactions and as a function of non-Kitaev couplings (Figs 2, 3) have been discussed in a few previous works, some of them cited in the manuscript, for instance Hickey et al. Nat. Comm.10, 530 (2019).

3) The magnetic field angle dependence of specific heat in the zigzag phase and in the Kitaev quantum spin liquid state have been studied and discussed in Tanaka et al. arXiv:2007.06757

Summary of Changes

Main text: We substantially revise the sections of Results and Discussion. Concrete results with extensive additional calculations are obtained, presented in the new figures (Figs. 2, 3, 4).

- In Fig. 1, we add a schematic figure (Fig. 1d) to emphasize our predictions in specific heat experiments with out-of-plane magnetic fields.
- The subsection “Topologically-protected critical lines and chirality” in the previous manuscript is split into the two subsections: “ Topologically protected critical lines” and “Chirality operator”.
- Results of the parton-based perturbation theory in the main-text of the previous manuscript are moved to Supplementary Information (Note 5).
- In the subsection of “Exact diagonalization”, the new figures (Figs. 2, 3) are presented to illustrate our main results with the chirality operator.
- Table 1 is added to summarize the parameter sets used in our calculations.
- Additional spin wave calculations are presented in Fig. 4.
- We modify Fig. 5 for a better illustration.
- In the Method section, details of the ED calculations and the phase diagrams are discussed, moving the contents of the previous manuscript to Supplementary Information.

Supplementary Information: We provide ten Supplementary Notes. Three of them (Notes 6, 7, 10) are newly added with additional calculations, and the others are from the Method section of the previous manuscript.

Reply to Reviewer #1

Reviewer: In this manuscript, the authors propose the magnetic field angle dependence of critical behaviors as an alternative identification for the Kitaev quantum spin liquid. The studies are based on the Kitaev model with additional Heisenberg and off-diagonal-symmetric interactions. The angle plane's critical line is topologically protected and intimately related to a chiral operator by symmetry arguments. Results from the parton theory and the exact diagonalization method also provide support for the proposal.

Authors: We thank the reviewer for an in-depth review of our manuscript.

R: However, this referee believes that the current manuscript shows insufficient scientific rigor and completeness in several aspects and cannot recommend publication:

A: Preparing the reply to the reviewers' reports, we realize that our presentation in the previous manuscript was not of the best form to convey scientific rigor and completeness of our results. We are grateful for the reviewer's comment which allows us to improve our manuscript. In the revised manuscript, we change the sections of Results, Discussion, and Method significantly to present our results better. Below, we provide our point-by-point answers to the reviewer's comments, and we believe that the revised version is now with sufficient scientific rigor and completeness.

R: (1) The study does not show the behavior of the real excitation gap, except for the pure Kitaev model. The Δ_{ED} in Fig. 4 is actually the absolute value of χ_{ED} (which is redundant, and the symbol is quite misleading, especially when Fig. 1 with a similar plot scheme features the parton gap - indeed an energy gap). Does the real gap show critical features aligning with the figures on the left? It is also a directly available result in both parton-based perturbation theory and exact diagonalization calculations.

A: We thank the reviewer for this critical comment. The reviewer is right that the chirality and its absolute value contain redundant information. In terms of the real excitation gap, it is unfortunate that Majorana energy gap values are only available from the parton-based perturbation theory near the pure Kitaev limit, and some of the results are illustrated in Fig. R1. The parton results show perfect agreements between the chirality and the energy gap.

FIG. R1. **Results of the parton-based perturbation theory.** Color maps of the chirality χ_{parton} (upper panels), the Chern number ν_{parton} (middle panels), and the Majorana energy gap Δ_{parton} (lower panels) on the plane of the field angles (θ, ϕ) . The parameter sets used in the four cases ($\#1 \sim 4$) are listed in Table 1 of the main text, and the magnetic field strength is fixed by $h = 0.01$ (horizontal axis: ϕ [°], vertical axis: θ [°]).

On the other hand, the exact diagonalization (ED) cannot provide the Majorana energy gap since spin excitations are always associated with additional flux excitations. In the pure Kitaev model as an example, a local spin-flip leads to the creation of one Majorana fermion (gapless) and two fluxes (gapped) [R1]. The spin excitation gap corresponds to the two-flux gap as shown by the exact calculations of the dynamical spin structure factor in Ref. [R2]. This can be also demonstrated by the exact diagonalization. Figure R2 shows the ED energy spectrum of the pure Kitaev model on a 24-site cluster, illustrating that the excitation gap of the ED calculations, Δ_{ED} , does not correspond to the energy of a single Majorana fermion. Furthermore, we compare Δ_{ED} with the ED calculations of the chirality, χ_{ED} , in Fig. R3. Again, the energy gap is not correlated with the sign change of the chirality due to the flux excitations that are always accompanying Majorana fermions.

We emphasize that the difficulty in extracting the excitation gap of a single Majorana fermion is a generic issue in most of spin-based calculations including ED and density matrix renormalization group (DMRG). With this motivation, we propose that the chirality allows us to investigate critical behaviors of the Majorana energy gap from the ground state wave function for the KQSL with D_3 point group symmetry.

In the revised manuscript, we remove the redundant results of the absolute value of the chirality

FIG. R2. **ED energy spectrum of the pure Kitaev model on a 24-site cluster.** **a** Lowest three energy levels of the pure Kitaev model ($K = -1$). Numbers in the parentheses indicate the degeneracy in each level. The ground states (flux-free) have three-fold topological degeneracy while the excited states (four-flux and two-flux) have degeneracies by translation and rotation symmetries. **b,c,d** Illustration of the flux patterns of the ground and excited states. Plaquettes with non-zero flux, $\langle \hat{W}_p \rangle = -1$, are marked by yellow. The gray lines depict the periodic boundary condition for the 24-site cluster, and the blue lines denote the bonds with $u_{jk} = -1$ in the associated parton Hamiltonian $H_{\text{eff}} = \sum_{\langle jk \rangle_\gamma} \frac{-iK}{4} u_{jk} c_j c_k$ with the same lattice geometry.

as the reviewer pointed out. Instead, we strengthen the discussion about the chirality as follows. We highlight the two universal features of the Majorana energy gap Δ_M :

1. **Symmetric zeroes:** for bond direction fields, topological transition/gap closing is guaranteed to occur by the symmetry, *i.e.*, $\Delta_M(\mathbf{h}) = 0$ for $(\theta = 90^\circ, \phi = n \cdot 60^\circ)$.
2. **Cubic dependence:** for in-plane fields, the h -cubic term governs low field behaviors of the KQSL, *e.g.*, $\nu_M(\mathbf{h}) \sim \text{sgn}(h_x h_y h_z)$ & $\Delta_M(\mathbf{h}) \sim |h_x h_y h_z|$ for $\theta = 90^\circ$.

Note that the two features arise due to the D_3 point group symmetry of the KQSL, irrespective of the microscopic details of spin interactions.

Most strikingly, we find that the chirality χ_{ED} shows the two features without using any Majorana descriptions. This new result is illustrated in Figs. 2, 3 of the revised manuscript. For example, the perfect match of $\text{sgn}(\chi_{\text{ED}})$ with ν_M is demonstrated for the pure Kitaev model in Fig. 2a. As a sanity check, we examine other A_2 spin operators up to the fifth order, which may contribute to the topological invariant (Chern number ν_M) of the KQSL, and find that the chirality is at least an order of magnitude larger than the expectation values of other A_2 spin operators.

All these results show that the topological properties of the KQSL are encoded in the expectation value of the chirality operator, which allows us to investigate the critical lines and other topological properties of the KQSL without using the parton descriptions.

R: (2) What are the scenarios of χ_{ED} and the real gap outside the Kitaev quantum spin liquid region? For instance, the authors illustrate the critical lines in various circumstances in the Kitaev

FIG. R3. **Chirality and the excitation gap from the exact diagonalization.** The field angle dependence of the chirality χ_{ED} and the gap Δ_{ED} is obtained for the pure Kitaev model with $K = -1$ and $h = 0.01$ (horizontal axis: ϕ [$^\circ$], vertical axis: θ [$^\circ$]).

quantum spin liquid region in Fig. 4. However, how do the critical lines behave in other phases? Such contrast is essential to establish the magnetic field angle dependence as a defining characteristic and intrinsic feature.

A: We appreciate the thoughtful comment that helps improve our manuscript. It is important to note that there is no critical line in magnetically ordered phases (under applied magnetic fields). Anisotropic spin interactions of the Kitaev and Gamma terms and the Zeeman coupling break all continuous spin rotation symmetries, so there is no gapless spin excitation. The presence of any critical lines implies an instability of the magnetic order by magnon condensation. In Fig. R4, we illustrate the chirality and the magnon gap for various magnetically ordered phases. Since ED calculations always give symmetric ground states, we employ a spin wave theory to calculate the chirality and the magnon gap. It is shown that the magnon gap does not have any critical lines, and obviously the magnon gap and the chirality do not show any resemblance/correlation at all.

In the revised manuscript, we add a few words to emphasize (i) the absence of critical lines in magnetically ordered phases and (ii) the absence of any correlation between the chirality and the magnon gap.

R: (3) The need to introduce the chiral spin operator is not clear when the gap value's result should be readily available. By symmetry argument, the chiral spin operator's expectation value vanishes where the gap vanishes, yet their proportionality is not guaranteed and only valid near the critical lines. Why forego the gap for a poor man's alternative that is the chiral spin operator?

FIG. R4. **Results of the spin wave theory for magnetically ordered phases.** Color maps of the chirality χ_{SW} (upper panels) and the magnon energy gap Δ_{SW} (lower panels) on the plane of the field angles (θ, ϕ) . The parameter sets used in the five cases ($\#5 \sim 9$) are listed in Table 1. The magnetic field strength is fixed by $h = 0.1$ in the cases $\#5 \sim 8$ and $h = 0.06$ in the case $\#9$. The black regions in the magnon gap indicate the regions where each magnetic order becomes unstable by magnon condensation (horizontal axis: ϕ [$^\circ$], vertical axis: θ [$^\circ$]).

A: As the reviewer points out, the chiral operator is not necessary if Majorana gap values are known. However, it is generally a challenging task to obtain the Majorana energy gap in numerical calculations including ED and DMRG, as explained in our reply to the reviewer's comment (1). We believe that our reply to the comment (1) faithfully answers this question. We appreciate the helpful comment.

R: (4) Using the gap value also helps the analysis be more quantitative and throughout, as it determines the temperature threshold the features smear out.

A: We believe that this comment is answered by our reply to the comment (1).

R: Besides, what role does the strength of the magnetic field h play?

A: We thank the reviewer for this nice question, which we find helpful for revising our manuscript. In Figs. 2,3 of the revised manuscript, we demonstrate the role of magnetic field strength by showing the shapes of the critical lines at various field strengths. The field evolutions of the critical lines can be understood by the competition between the h -cubic and h -linear dependences, which is a consequence of the interplay between non-Kitaev interactions and the magnetic field.

R: What is the optimal magnetic field strength?

A: We are afraid if we misunderstand this question, so we consider three possible cases. First, if the question is about the optimal strength to realize the non-abelian KQSL, it has been discussed in literature such as Hickey et al. Nat. Comm. 10, 530 (2019). Second, if the question is about the optimal strength to observe our proposed field angle dependence, we find that any field strengths which do not destabilize the KQSL show the characteristic field angle dependence. Third, if the question is about real materials such as α -RuCl₃, the field range to stabilize the KQSL is reported to be 7~12 T [R3–R6]. The microscopic model Hamiltonian for α -RuCl₃ has been hotly debated with no consensus yet [R7]. Instead of finding an accurate model Hamiltonian for the material system, we consider a generic Hamiltonian and focus on characteristics of the non-abelian KQSL via the magnetic field angle dependence.

R: The manuscript only shows results for a select parameter setting at the very end and largely insufficient considering the authors are making experimentally realistic proposals for alternative KQSL features.

A: We have performed extensive calculations for a variety of parameter sets, and we present the results in the revised manuscript. The ED calculations of the chirality are presented in Figs. 2, 3 and Supplementary Fig. 7, and the results of the parton-based perturbation theory and the spin wave theory are provided in Fig. 4 and Supplementary Figs. 2, 4. The characteristic zero lines of the chirality and the two universal features (symmetric zero and cubic dependence) are confirmed in all the cases of the KQSL.

We would like to stress that our results can be directly applicable to experiments, and the two universal features have been recently observed in the experimental work by Tanaka et al. [R6]. Beyond the experiment, our work further characterizes the effects of non-Kitaev interactions. As an example, it is demonstrated how the specific heat measurements for the critical lines are modified by non-Kitaev interactions, suggesting a new pathway to quantitatively characterize non-Kitaev interactions from experimentally measured critical lines. We expect that our calculations will be a useful guide for future theoretical and experimental studies on the critical lines and other topological properties of the KQSL.

R: In total, this manuscript begins with an interesting idea yet falls short in subsequent justifica-

tions and illustrations and is not suitable for academic publication in its current form.

A: We believe that we have addressed all the issues raised by the reviewer. With these revisions, we hope that the reviewer finds our paper is now suitable for publication in Nature Communications.

-
- [R1] Baskaran, G., Mandal, S., & Shankar, R. Exact results for spin dynamics and fractionalization in the Kitaev model. *Phys. Rev. Lett.* **98**, 247201 (2007).
- [R2] Knolle, J., Kovrizhin, D. L., Chalker, J. T., & Moessner, R. Dynamics of a two-dimensional quantum spin liquid: Signatures of emergent Majorana fermions and fluxes. *Phys. Rev. Lett.* **112**, 207203 (2014).
- [R3] Sears, J. A., Zhao, Y., Xu, Z., Lynn, J. W. & Kim, Y.-J. Phase diagram of α -RuCl₃ in an in-plane magnetic field. *Phys. Rev. B* **95**, 180411 (2017).
- [R4] Banerjee, A. et al. Excitations in the field-induced quantum spin liquid state of α -RuCl₃. *npj Quantum Mater.* **3**, 8 (2018).
- [R5] Kasahara, Y. et al. Majorana quantization and half-integer thermal quantum Hall effect in a Kitaev spin liquid. *Nature (London)* **559**, 227 (2018).
- [R6] Tanaka, O. et al. Thermodynamic evidence for field-angle dependent Majorana gap in a Kitaev spin liquid. Preprint at <https://arxiv.org/abs/2007.06757> (2020).
- [R7] Laurell, P. & Okamoto, S. Dynamical and thermal magnetic properties of the Kitaev spin liquid candidate α -RuCl₃. *npj Quantum Mater.* **5**, 2 (2020).

Reply to Reviewer #3

Reviewer: Hwang et al. present a theoretical study of the ground state of the Kitaev model with inclusion of non-Kitaev terms (Heisenberg J and the so-called Gamma and Gamma prime terms) in the absence and presence of a magnetic field. For that they use parton theory in the region where the non-Kitaev terms can be treated perturbatively, as well as exact diagonalization of 24-site clusters, which allow to extend the study to finite values of J, Gamma and Gamma prime. The work is specially devoted to analyze the Chern number of the magnetic field induced phases as a function of the magnetic field angle and to the evaluation of such signatures in thermodynamic quantities such as specific heat. This is a well-done study that presents nice results worth publishing.

A: We are grateful to the reviewer for the thorough summary of our work and are happy to see the comment, *“This is a well-done study that presents nice results worth publishing.”*

I’m less convinced that the results are of enough novelty to have the work published as a Nature Communications. The reasoning is the following:

A: Though we believe our results are novel, we also admit that our presentation in the previous manuscript was not of the best form. We appreciate the reviewer’s comment and we perform a major revision of the manuscript.

The novelty of this work is manifested by our two claims based on the symmetries of time reversal and D_3 point group. First, we show that the critical lines must exist on the field angle plane due to the topological nature of the non-abelian Kitaev quantum spin liquid (KQSL). We propose the critical lines as a measurable smoking gun signature of the KQSL. Second, we also show that the topological properties of the KQSL are encoded in the expectation value of the chirality operator, which allows us to investigate the critical lines and other topological properties of the KQSL without using parton descriptions. Our work also demonstrates how the critical lines are affected by non-Kitaev interactions, suggesting a new route to quantitatively characterize non-Kitaev interactions from experimentally measured critical lines.

We believe that the current version reveals the novelty of our work much better, hopefully ready to be published in Nature Communications. Below, we provide our point-by-point replies to the reviewer’s comments.

R: (1) Calculations of the Chern number (Fig. 1) for the Kitaev model under various applied field directions have been discussed previously, see for instance Yokoi et al. arXiv:2001.01899.

A: The reviewer is correct that the Chern number calculations in Fig. 1 have been discussed in literature, including the preprint by Yokoi et al. (arXiv:2001.01899) and even the original paper by Kitaev. We do not claim that the Chern number calculations in Fig.1 are new, and instead, our intention is to compare the Chern number calculations with the evaluations of the chirality operator by exact diagonalization (ED) as shown in Fig. 2a of the revised manuscript. We find a perfect agreement between the ED calculations and the parton analysis of the Chern number, which allows us to use the chirality operator to investigate topological properties of the non-abelian KQSL. Based on the perfect match, we extend the application of the chirality operator to more general cases with non-Kitaev spin interactions including Γ , Γ' , and Heisenberg terms and uncover how non-Kitaev interactions affect the critical lines and related topological properties as shown in Figs. 2, 3 of the revised manuscript.

R: (2) The variety of phases obtained from exact diagonalization for ferromagnetic and antiferromagnetic Kitaev interactions and as a function of non-Kitaev couplings (Figs 2, 3) have been discussed in a few previous works, some of them cited in the manuscript, for instance Hickey et al. Nat. Comm. 10, 530 (2019).

A: The reviewer is correct that the various phases have been discussed in previous literatures such as Hickey et al. Nat. Comm. 10, 530 (2019). The phase diagrams in Figs. 2, 3 were intended to emphasize different roles of non-Kitaev interactions in destabilizing the ferromagnetic KQSL and the antiferromagnetic KQSL. Still, we agree with the reviewer that the phase diagrams do not show enough novelty, so we move the phase diagrams to the Method section, where we provide details of our ED calculations.

In the revised manuscript, we instead focus on our two main claims about critical lines and chirality operator, and in Figs. 2, 3 we present ED calculations of the chirality as supporting evidences for our claims.

R: (3) The magnetic field angle dependence of specific heat in the zigzag phase and in the Kitaev quantum spin liquid state have been studied and discussed in Tanaka et al. arXiv:2007.06757.

A: We thank the reviewer to bring up the similarity to the experimental work by Tanaka et al. (arXiv:2007.06757). In fact, our theoretical study has been motivated by the experiment, and the theory and experiment have been actively communicating with each other for a clear identification of the non-abelian KQSL phase in α -RuCl₃ (three authors in this study are also co-authors of the experimental work). Our theoretical calculations are shown to be fully consistent with the actual measurements of the specific heat for the KQSL and zigzag antiferromagnetic phases.

More importantly, our theoretical study provides further useful suggestions and guidelines for future experiments. In Fig. 1d of the revised manuscript, we visualize how the field angle dependence measurements of specific heat can reveal the critical lines that we propose as a smoking gun signature of the KQSL. Furthermore, our results in Figs. 2, 3 clearly illustrate how the specific heat measurements for the critical lines are modified by non-Kitaev interactions. These results can be used as a new guideline to experimentally evaluate the relative strength of Kitaev and non-Kitaev interactions from the angle dependence measurements of specific heat.

We believe that the revised manuscript reveals the novelty of our work much better. We hope the reviewer finds our paper is suitable for publication in Nature Communications.

REVIEWER COMMENTS

Reviewer #1 (Remarks to the Author):

The authors' revised manuscript is quite an improvement over the previous version. Also, their response to my previous report has partially cleared out my concerns and made their motivations clearer. Overall, if all conclusions are correct, this manuscript offers smoking-gun signatures to the so-far elusive KQSL and has sufficient novelty and potential impact for publication.

However, this referee still thinks some disconnections between the reasoning need to be bridged and is therefore not fully convinced. Until such lingering confusion and doubts are cleared out, this referee feels unconfident to recommend this manuscript for Nature Communications.

(1) The authors explained in their response that the Majorana energy gap is hard to get. However, this referee feels that the spectrum gap is more vital, which should be available in ED calculations. The reason is the following: which gap does the heat capacity (e.g., Fig. 5), the experimental criteria the authors proposed as a smoking-gun signature, depend on, the Majorana energy gap or the spectrum gap? This referee thinks it should be the latter, especially when we need to compare KQSL with the non-KQSL phases whose heat capacity is primarily contributed by the magnon excitations instead.

(2) The authors showed that the chirality χ (chiral operator expectation value) is a signature quantity for the topological invariant, the critical lines, and, importantly, the gap with an unusual magnetic field dependence h^3 . On the other hand, the correspondence between the chirality and the gap is only established in or near the pure Kitaev limit ($K \gg \Gamma, \Gamma'$). Is there any evidence or reason it holds more generally in the rest of the KQSL phase diagram? This is important since the following discussions, including the magnetic field dependence (h^3 instead of h , essential for establishing the difference in the heat capacity behaviors), are based upon the behavior of the chirality.

In other words, while $\text{gap}_{\text{parton}}$ scales as h^3 in the pure Kitaev limit and $\text{gap}_{\text{magnon}}$ scales as h for non-KQSL phases (Fig. 5), the gap versus h scaling for KQSL further away from the pure Kitaev limit is solely based upon the chirality χ (Fig. 2 and Fig. 3). However, the connection between χ and gap (mass) is only established near the pure Kitaev limit. Therefore, it is not firmly established whether the gap behaves similarly to χ for the scenarios in Fig. 2 and Fig. 3.

Overall, while this referee believes the authors' results are solidly grounded for KQSL near the pure Kitaev limit, there still seem to be loopholes for both numerical (#1) and theoretical (#2) arguments of the proposed gap and specific heat behaviors for KQSL further away from the pure Kitaev limit.

Reviewer #3 (Remarks to the Author):

The resubmitted version of the manuscript by Hwang et al. focuses now on two aspects:

- Analysis of the topology of magnetic field induced phases (as a function of the magnetic field angle) of the Kitaev model through topological invariants by exploiting time reversal invariance and D3 symmetry.

- The definition and performance of a chirality operator (eq. 4) to identify the topological nature of magnetic field induced phases in the Kitaev model and extensions of the model to non-Kitaev terms as a function of the magnetic field angle.

For that the authors perform exact diagonalization calculations on 24-site clusters.

The structure of the manuscript improved with respect to the previous one and the authors clarified most of the issues. There are however some open questions left:

(1) the authors show in Fig. 2 that the chirality may not be consistent with the Chern number

when doing calculations up to third order perturbation with the partonic representation. How is this discrepancy to be interpreted?

is this a limitation of the perturbation theory and parton method, or is it showing the limits of using the chirality operator as an indicator for topological phases?

this has important implications since the authors suggest to use the chiral operator as a topology indicator.

(2) how suitable is to use the chirality operator to identify distinct phase boundaries under magnetic field?

After clarification of these issues, the manuscript could be suitable for publication.

Reply to Reviewer #1

Reviewer: The authors' revised manuscript is quite an improvement over the previous version. Also, their response to my previous report has partially cleared out my concerns and made their motivations clearer. Overall, if all conclusions are correct, this manuscript offers smoking-gun signatures to the so-far elusive KQSL and has sufficient novelty and potential impact for publication.

Authors: We thank the reviewer for a thorough review of our manuscript once again. Also, we are glad to see the reviewer's comment "*if all conclusions are correct, this manuscript offers smoking-gun signatures to the so-far elusive KQSL and has sufficient novelty and potential impact for publication.*"

R: However, this referee still thinks some disconnections between the reasoning need to be bridged and is therefore not fully convinced. Until such lingering confusion and doubts are cleared out, this referee feels unconfident to recommend this manuscript for Nature Communications.

A: We appreciate the critical comments which allow us to improve our manuscript further. We provide our point-by-point responses to the reviewer's comments below.

R: (1) The authors explained in their response that the Majorana energy gap is hard to get. However, this referee feels that the spectrum gap is more vital, which should be available in ED calculations. The reason is the following: which gap does the heat capacity (e.g., Fig. 5), the experimental criteria the authors proposed as a smoking-gun signature, depend on, the Majorana energy gap or the spectrum gap? This referee thinks it should be the latter, especially when we need to compare KQSL with the non-KQSL phases whose heat capacity is primarily contributed by the magnon excitations instead.

A: We fully agree with the reviewer that the spectrum gap determines heat capacity at low temperatures, which can be obtained by ED calculations, in principle. Yet, we would like to point out large finite-size effects in energy spectrum, which make it difficult to connect our ED results on a 24-site system ($N_s = 24$) to the energy gap structure in the thermodynamic limit ($N_s = \infty$).

We analyze the pure Kitaev model in different system sizes to understand the finite-size effects further. Figure R1(a) illustrates the system size dependence of the Majorana gap $\Delta_{\text{parton}}(N_s)$

FIG. R1. **Finite-size effects in the energy spectrum of the zero-field pure Kitaev model.** (a) Majorana energy gap Δ_{parton} as a function of $1/\sqrt{N_s}$ (N_s : number of sites). Δ_{parton} is obtained by the parton theory for $N_s = 24L^2$ ($L = 1, 2, \dots, 10$). The flux-free sector is considered, where the ground state appears. Dashed line: the flux gap $\Delta_{\text{flux}} = 0.065$ in the thermodynamic limit. (b) Parton (Majorana fermion) energy levels in the flux-free sector of the 24-site system. (c) ED energy spectrum of the 24-site system. In the flux-free sector, the excitation gap is exactly given by $2\Delta_{\text{parton}} = 2 \times 0.366$, thus a two-Majorana excitation. Flux gap is given by $\Delta_{4F} = 0.034$. In these calculations, the Kitaev coupling is fixed by $K = -1$.

from the parton analysis. It is clear that the Majorana gap becomes zero in the thermodynamic limit; $\Delta_{\text{parton}}(N_s \rightarrow \infty) = 0$. Figure R1(b) shows the parton (Majorana fermion) energy levels of the 24-site system, where the Majorana gap is $\Delta_{\text{parton}}(N_s = 24) = 0.366$. In Fig. R1(c), the ED energy spectrum for the same system size is presented. In each flux sector, only a few lowest energy levels are specified (many other excited states are dropped for a clear readability of the figure).

Several important points are emphasized from the above results. First, the exact solvability of the pure Kitaev model allows us to find the one-to-one correspondence between the parton analysis and the ED calculations. The ED energy spectrum can be reproduced by occupying the parton energy levels with Majorana fermions. Second, the ED calculations only find (gauge-invariant) physical excitations such as pairs of Majorana fermions whose minimum energy is $2\Delta_{\text{parton}} = 0.732$ for the 24-site system [Fig. R1(c)]. Third, there exist other types of states below the minimum energy $2\Delta_{\text{parton}}$. For example, the first excited state corresponds to the four-flux-created state with the energy

$$\Delta_{4F} = 0.034 \ll 2\Delta_{\text{parton}}(N_s = 24) = 0.732, \quad (\text{R1})$$

indicating that the energy scale $2\Delta_{\text{parton}}(N_s = 24)$ is huge and many other states exist below the

scale. On the other hand, the thermodynamic limit results are known to be

$$\Delta_{\text{flux}} = 0.065 \gg 2\Delta_{\text{parton}}(N_s \rightarrow \infty) = 0. \quad (\text{R2})$$

The delocalized Majorana excitations are quite sensitive to the system size unlike the flux excitations.

Based on the results, we conclude that the 24-site ED calculations cannot be used to determine $\Delta_{\text{parton}}(N_s \rightarrow \infty)$ and refrain from using the ED spectrum gap to predict behaviors of physical quantities in the thermodynamic limit such as specific heat, even though we fully agree with the reviewer that the spectrum gap is vital. We thank the reviewer for the thoughtful comment.

R: (2) The authors showed that the chirality χ (chiral operator expectation value) is a signature quantity for the topological invariant, the critical lines, and, importantly, the gap with an unusual magnetic field dependence h^3 . On the other hand, the correspondence between the chirality and the gap is only established in or near the pure Kitaev limit ($K \gg \Gamma, \Gamma'$). Is there any evidence or reason it holds more generally in the rest of the KQSL phase diagram? This is important since the following discussions, including the magnetic field dependence (h^3 instead of h , essential for establishing the difference in the heat capacity behaviors), are based upon the behavior of the chirality. In other words, while Δ_{parton} scales as h^3 in the pure Kitaev limit and Δ_{magnon} scales as h for non-KQSL phases (Fig. 5), the gap versus h scaling for KQSL further away from the pure Kitaev limit is solely based upon the chirality χ (Fig. 2 and Fig. 3). However, the connection between χ and gap (mass) is only established near the pure Kitaev limit. Therefore, it is not firmly established whether the gap behaves similarly to χ for the scenarios in Fig. 2 and Fig. 3.

A: We appreciate the reviewer's invaluable comment which helps us to refine our arguments. As the reviewer correctly pointed out, the relation between the chirality (χ) and the Majorana energy gap (Δ) has been mainly established by using the perturbative parton analysis near the pure Kitaev model. We would like to point out that proving the relation for general cases is extremely difficult. Nevertheless, we believe that the relation holds generally in our KQSL phase diagrams and provide two additional evidences below.

First, we find that the relation $|\chi| \propto \Delta$ holds near the universal zeroes for a *generic* KQSL with D_3 symmetry. To show this, let us recall that magnetic fields along the bond directions do not break the bond direction C_2 symmetry, which makes both χ and Δ to be zero (properties of the

FIG. R2. **Chirality of the model H_χ .** The chirality χ as a function of the coupling constant M . Red: ED calculation results for the 24-site cluster. Blue: parton analysis results for the infinite system. In all these results, the Kitaev coupling is fixed by $K = -1$.

universal zeroes as discussed in the manuscript). Suppose we slightly tilt the magnetic field from a bond direction by a small angle $\delta\phi$ ($\ll 1$). The tilting effects appear in the parton Hamiltonian with the notation of the original Kitaev model as

$$\begin{pmatrix} 0 & iU_{\mathbf{q}} \\ -iU_{\mathbf{q}}^* & 0 \end{pmatrix} \xrightarrow{\text{Tilting the field by } \delta\phi} \begin{pmatrix} m\delta\phi & iU_{\mathbf{q}} \\ -iU_{\mathbf{q}}^* & -m\delta\phi \end{pmatrix}, \quad (\text{R3})$$

with a generic form of $U_{\mathbf{q}}$. Then, the mass generation of Majorana fermions ($\Delta = |m\delta\phi|$) is obvious, indicating that the mass term is in the A_2 representation. Then, both of χ and Δ are proportional to $\delta\phi$ because both of them are in the A_2 representation, and the linear relationship, $|\chi| \propto \Delta$, is thus established near the universal zeroes of a generic KQSL. We stress that the D_3 symmetry plays a significant role in our results. In other words, the universal zeroes always appear in a generic KQSL with the D_3 symmetry, and the relationship must hold in any points of KQSL in our phase diagrams.

Second, we consider another exactly solvable model,

$$H_\chi = K \sum_{\langle jk \rangle_\gamma} S_j^\gamma S_k^\gamma - M \sum_p \hat{\chi}_p, \quad (\text{R4})$$

which consists of the Kitaev interactions (K) and the chirality operator three-spin interactions (M). The exact parton analysis shows that the Chern number is $\nu = \text{sgn}(M)$ and the Majorana gap is $\Delta = \frac{3\sqrt{3}}{4}|M|$. Figure R2 presents the chirality χ as a function of M with ED and parton calculations. The positive correlation between χ and M is demonstrated, establishing the connection between the chirality and the Majorana gap in a fairly large window ($|M/K| \lesssim 0.5$).

In the revised manuscript, we add a few words about the two evidences in the main-text and SI, emphasizing the special role of the universal zeroes in the relation, $|\chi| \propto \Delta$. We are grateful to the reviewer for the invaluable comment.

R: Overall, while this referee believes the authors' results are solidly grounded for KQSL near the pure Kitaev limit, there still seem to be loopholes for both numerical (#1) and theoretical (#2) arguments of the proposed gap and specific heat behaviors for KQSL further away from the pure Kitaev limit.

A: We believe that we have addressed all the issues raised by the reviewer. We hope that the reviewer is convinced now and finds our paper suitable for publication in Nature Communications.

Reply to Reviewer #3

Reviewer: The resubmitted version of the manuscript by Hwang et al. focuses now on two aspects:

- Analysis of the topology of magnetic field induced phases (as a function of the magnetic field angle) of the Kitaev model through topological invariants by exploiting time reversal invariance and D_3 symmetry.
- The definition and performance of a chirality operator (Eq. 4) to identify the topological nature of magnetic field induced phases in the Kitaev model and extensions of the model to non-Kitaev terms as a function of the magnetic field angle. For that the authors perform exact diagonalization calculations on 24-site clusters.

Authors: We thank the reviewer for the comprehensive review once again.

The structure of the manuscript improved with respect to the previous one and the authors clarified most of the issues. There are however some open questions left:

A: It is nice to see the reviewer's comment "*The structure of the manuscript improved with respect to the previous one and the authors clarified most of the issues*". We answer the reviewer's questions below.

R: (1) The authors show in Fig. 2 that the chirality may not be consistent with the Chern number when doing calculations up to third order perturbation with the partonic representation. How is this discrepancy to be interpreted? Is this a limitation of the perturbation theory and parton method, or is it showing the limits of using the chirality operator as an indicator for topological phases? This has important implications since the authors suggest to use the chiral operator as a topology indicator.

A: We thank the reviewer for bringing up this important issue. As the reviewer mentioned, the perturbative parton theory up to the third order and the chirality operator method with ED calculations show discrepancy for the cases (#3,4) in Fig. 2 of the main-text. The discrepancy should be interpreted as the signal that additional careful analysis is necessary.

The two methods have the limit where they become exact, the pure Kitaev model under weak magnetic field. But, both of the two methods have their own validity conditions in practice. It

FIG. R3. **Expectation values of the A_2 operators.** Left: pictorial illustration of the operators $\hat{\chi}_{3,5,7}$. Right: the expectation values $\langle \hat{\chi}_{3,5,7} \rangle$ for the four cases #1,2,3,4. The y-axis is in a log scale, and all the results are obtained by 24-site ED calculations with the field $h = 0.01$ || [111].

is exceedingly difficult to prove which one is a better approach in general. Additional careful analysis for a better judgement is necessary if the two methods show discrepancy. Below, we provide the information on the validity conditions of the two methods and discuss how to resolve the discrepancy.

Recall that the topological invariant (ν) is determined by the zero temperature limit of thermal hall conductivity over temperature, κ_{xy}/T . The perturbative parton theory introduces parameters made of non-Kitaev interactions and flux gap such as $\Gamma/\Delta_{\text{flux}}$, and the topological invariant can be written as the expansion,

$$\nu = \text{Sign}[m_0 + m_1\left(\frac{\Gamma}{\Delta_{\text{flux}}}\right) + m_2\left(\frac{\Gamma}{\Delta_{\text{flux}}}\right)^2 + \dots]. \quad (\text{R5})$$

The coefficients $m_{0,1,\dots}$ are functions of the coupling constants of a given Hamiltonian ($K, J, \Gamma, \Gamma', \dots$). The perturbative parton theory is valid if the parameters such as $\Gamma/\Delta_{\text{flux}}$ are small enough. The case #2 is expected to be similar to the case #1 as manifested in Fig. 2 of the main-text. For the cases (#3,#4), the perturbation parameters are larger, for example $\Gamma/\Delta_{\text{flux}} = 0.77$. Though the perturbative parton theory is powerful, we also note that the convergence of the expansion is neither guaranteed nor proven, especially with multiple coupling constants.

The chirality operator method, on the other hand, is based on a different type of expansions by exploiting symmetry properties. Since ν is in the time reversal odd A_2 representation of D_3 symmetry, we consider the operator associated with κ_{xy} , commutator/correlator of energy currents as used in [PRL 119,127204], which can be formally written as

$$\hat{A}_2 = M_3 \sum \hat{\chi}_3 + M_5 \sum \hat{\chi}_5 + M_7 \sum \hat{\chi}_7 + \dots \quad (\text{R6})$$

The expectation values ($\langle \hat{\chi}_{3,5,7} \rangle$) determine ν by the fluctuation-dissipation theorem. The string operators with an odd number of spin operators connecting two sites on a same sublattice ($\hat{\chi}_{3,5,7}$) are introduced, whose graphical representations are given in Fig. R3. As explained in Supplementary Information, there is no linear term in the expansion, and the first term $\sum \hat{\chi}_3$ is identical to the chirality operator $\sum_p \hat{\chi}_p$. The coefficients $M_{3,5,7}$ are functions of the coupling constants of a given Hamiltonian ($K, J, \Gamma, \Gamma', \dots$) while the string operators are independent. The chirality operator method is to truncate the expansion at the leading chirality operator term to determine ν .

The ED calculations with the chirality operator are valid under the conditions,

$$\left| M_3 \sum \langle \hat{\chi}_3 \rangle \right| \gg \left| M_5 \sum \langle \hat{\chi}_5 \rangle \right|, \left| M_7 \sum \langle \hat{\chi}_7 \rangle \right|, \dots \quad (\text{R7})$$

The conditions hold near the pure Kitaev model as manifested in the perfect matches between the ED calculations and the perturbative parton theory for the cases (#1, 2). The structure of the chirality operator method can be further analyzed by evaluating the expectation values of $\hat{\chi}_{3,5,7}$ for the cases (#1-4) with a magnetic field along the [111] direction, shown in Fig. R3. We find that the cases (#3,4) have hundred times larger values of $|\langle \hat{\chi}_n \rangle|$ compared to the cases (#1,2), which are consistent with the larger perturbation parameters of the perturbative parton theory. Moreover, we find that the two ratios,

$$r_1 \equiv \frac{|\langle \hat{\chi}_5 \rangle|}{|\langle \hat{\chi}_3 \rangle|} \sim \frac{1}{10}, \quad r_2 \equiv \frac{|\langle \hat{\chi}_7 \rangle|}{|\langle \hat{\chi}_3 \rangle|} \sim \frac{1}{100}$$

for the four cases. Since the chirality operator method works well for the cases (#1,2), the validity conditions, (R7), can be fulfilled if the coefficients $M_{3,5,7}$ of the cases (#3,4) are not much different from the ones of the cases (#1,2).

To check the validity conditions (R7) explicitly, one needs to calculate the coefficients $M_{3,5,7}$ from scratch. We believe that this problem is beyond the scope of the current work and leave it for future works. Instead, we notice that the coefficients $M_{3,5,7}$ are in the trivial representation of the D_3 symmetry, being functions of energy eigenvalues. We calculate the energy spectrum variances of the lowest hundred energy eigenvalues for the four cases and find that the variances are all less than 5%. Thus, it is tempting to assume that $\overline{M_{3,5,7, \dots}}$ do not vary much, and then, the ED calculations with the chirality operator would work.

We would like to stress that our proposal with the chirality operator is not only alternative but also complementary to the perturbative parton analysis. Namely, the perfect matches between the two methods' results for the cases (#1,2) become a sanity check as shown in Fig. 2 of the

FIG. R4. **Distinct phase boundaries in the chirality.** (a,b,c) ED results for $K = -1$, $\Gamma' = 0.05$, $h = 0.01 \parallel [111]$ with the flux W , the chirality χ , and a comparison of W and χ along the $J = 0$ line cut. (d,e,f) ED results for $K = 1$, $\Gamma' = -0.05$, $h = 0.01 \parallel [111]$ with the flux W , the chirality χ , and a comparison of W and χ along the $J = 0$ line cut.

main-text. If the two methods show discrepancy as in the cases (#3,4), further analysis of the systems are necessary. For the perturbative parton analysis, one obvious way is to perform higher order perturbations. For the chirality operator method, one can try other numerical methods such as density-matrix-renormalization-group (DMRG) calculations since ED calculations with a larger system size are numerically difficult. The topological invariant is believed to be less susceptible to slight symmetry breaking from a cylinder-like lattice shape, and the chirality operator method is expected to be useful even with DMRG calculations.

We also believe that there are plenty of rooms to extend and apply our results where new ideas and approaches can be developed. In the revised manuscript, we add a few words to discuss the above points in the main-text and SI. We are grateful to the reviewer for the invaluable comment.

R: (2) how suitable is to use the chirality operator to identify distinct phase boundaries under magnetic field?

A: The chirality operator is also applicable to identify distinct phase boundaries under magnetic fields, which makes the operator is a complementary tool to detect quantum phase transitions. We demonstrate this by conducting additional ED calculations as shown in Fig. R4. We find that the phase boundaries revealed by the plaquette operator (\hat{W}_p) are equally well captured by

the chirality operator ($\hat{\chi}_p$). In some cases, the chirality operator works better than the plaquette operator as shown in Fig R4b,e (marked by arrows). This shows that the chirality is also useful for the identification of distinct phase boundaries of the Kitaev system. We appreciate the reviewer's helpful comment and add one section in SI of the revised manuscript.

R: After clarification of these issues, the manuscript could be suitable for publication.

A: We believe we have clarified all the issues raised by the reviewer. With these revisions we hope that the reviewer finds our paper is now suitable for publication in Nature Communications.

REVIEWERS' COMMENTS

Reviewer #1 (Remarks to the Author):

The authors have responded to my previous concerns satisfactorily. The added data and discussions in the main text and supplemental materials show that the authors' approximations and compromises are justified. Therefore, this referee sees no further obstacle towards its acceptance for future publication.

The authors are suggested to pay more attention to their detailed presentations, though: "symmetric zeros" near the end of page 3 is clearly a broken sentence; the color scale in the top panels of Fig. 3(a) and 3(b) needs attention (too faint), as no color seems to appear on the printed manuscript (despite faint colors visible in the pdf on screen).

Reviewer #3 (Remarks to the Author):

The revised version of the manuscript has improved with respect to the previous one and the authors have answered the questions of the referees rather satisfactorily. While the idea of introducing the chirality operator to identify the topological nature of magnetic field induced phases in Kitaev models, is a good one, its use as a smoking gun quantity for models beyond the Kitaev model remains to be further tested. Nevertheless the authors make a case and connect it to the (magnetic field) angle-dependence of thermodynamic quantities such as specific heat. Bringing such a suggestion for discussion in the community may deserve publication.

Two issues that the authors should check:

1) On page 5, end of second column, the authors comment that the behaviors shown in Fig. 5 correspond to those observed in ref. 65. Comparing of Fig. 5 in the manuscript and Fig. 2 in arXiv:2007.06757 one observes that what is predicted for the QSL in Fig. 5a would correspond to the observations in α -RuCl₃ shown in Fig.2c in arXiv:2007.06757 which is the region where the system shows zigzag order. Could the authors clarify? ther seems to be an apparent problem?

2) actualize references in the manuscript.

There are various thermodynamic studies on these Kitaev materials analyzing the dependence of thermodynamic quantities on the magnetic field angle.

Reply to Reviewer #1

Reviewer (R): The authors have responded to my previous concerns satisfactorily. The added data and discussions in the main text and supplemental materials show that the authors' approximations and compromises are justified. Therefore, this referee sees no further obstacle towards its acceptance for future publication.

Authors (A): We are delighted to see reviewer's satisfaction in our reply and revision and glad that the reviewer agrees on the acceptance of our manuscript for publication in Nature Communications.

R: The authors are suggested to pay more attention to their detailed presentations, though: "symmetric zeros" near the end of page 3 is clearly a broken sentence; the color scale in the top panels of Fig. 3(a) and 3(b) needs attention (too faint), as no color seems to appear on the printed manuscript (despite faint colors visible in the pdf on screen).

A: We deeply appreciate the reviewer's efforts on reviewing our manuscript. In the revised manuscript, we fix the broken sentence in page 3 and change the color scheme of Fig. 3. We check the color is clearly visible both on printed and screen versions. We believe that our revised manuscript is ready to be published.

Reply to Reviewer #3

Reviewer: The revised version of the manuscript has improved with respect to the previous one and the authors have answered the questions of the referees rather satisfactorily. While the idea of introducing the chirality operator to identify the topological nature of magnetic field induced phases in Kitaev models, is a good one, its use as a smoking gun quantity for models beyond the Kitaev model remains to be further tested. Nevertheless the authors make a case and connect it to the (magnetic field) angle-dependence of thermodynamic quantities such as specific heat. Bringing such a suggestion for discussion in the community may deserve publication.

Authors: We are happy that the reviewer is satisfied with our revision and answers. It is very nice to see the reviewer's comment, "*Bringing such a suggestion for discussion in the community may deserve publication.*".

R: Two issues that the authors should check. (1) On page 5, end of second column, the authors comment that the behaviors shown in Fig. 5 correspond to those observed in Ref. 65. Comparing of Fig. 5 in the manuscript and Fig. 2 in arXiv:2007.06757 one observes that what is predicted for the KQSL in Fig. 5a would correspond to the observations in α -RuCl₃ shown in Fig. 2c in arXiv:2007.06757 which is the region where the system shows zigzag order. Could the authors clarify? There seems to be an apparent problem?

A: We truly appreciate the reviewer's comprehensive review and thoughtful comments once again. We would like to mention that our notation for the field angle ϕ is **different** from the one of Ref. 65 (arXiv:2007.06757) and our results on Kitaev quantum spin liquids (KQSL) are consistent with the interpretation of Ref. 65. In fact, three of the authors of our manuscript participated in Ref. 65.

To avoid any confusions from the different notations, we adopt the notation of Ref. 65, which is more widely used in community, and redraw all the figures in the revised manuscript. With the new notation, the field angle dependence becomes more apparent. Fig. 5a of our revised manuscript matches well with Fig. 2d of Ref. 65, where KQSL behaviors are expected, instead of Fig. 2c.

R: (2) Actualize references in the manuscript. There are various thermodynamic studies on these Kitaev materials analyzing the dependence of thermodynamic quantities on the magnetic field angle.

A: We thank the reviewer for letting us know the existence of the other works. In the revised manuscript, we have included several references about field angular dependence of thermodynamic quantities as follows.

- Modic, K.A. *et al.* Resonant torsion magnetometry in anisotropic quantum materials. *Nat. Commun.* 9, 3975 (2018).
- Riedl, K., Li, Y., Winter, S. M. & Valentí, R. Sawtooth Torque in Anisotropic $j_{\text{eff}} = 1/2$ Magnets: Application to α -RuCl₃. *Phys. Rev. Lett.* 122, 197202 (2019).
- Gordon, J. S. & Kee, H.-Y., Testing topological phase transitions in Kitaev materials under in-plane magnetic fields: Application to α -RuCl₃, *Phys. Rev. Research* 3, 013179 (2021).
- Bachus, S. *et al.* Angle-dependent thermodynamics of α -RuCl₃. *Phys. Rev. B* 103, 054440 (2021).